# Effects of Outdoor Stocking Density on Growth, Feather Damage and Behavior of Slow-Growing Free-Range Broilers

**DOI:** 10.3390/ani11030688

**Published:** 2021-03-04

**Authors:** Hannah N. Phillips, Bradley J. Heins

**Affiliations:** Department of Animal Science, University of Minnesota, 1364 Eckles Avenue, Saint Paul, MN 55108, USA

**Keywords:** outdoor stocking density, free-range, broilers, ranging, behavior, welfare

## Abstract

**Simple Summary:**

Providing meat-type chickens free access to pasture is intended to improve their wellbeing by allowing them opportunities to express their natural behaviors and mitigate discomfort associated with indoor housing. However, there is limited information on how much pasture space is needed to improve the quality of life for these chickens. In a controlled experiment, chickens with a larger amount of pasture space had reduced feather damage and showed less aggression towards other chickens. Depending on age, chickens with a smaller amount of pasture space did more stretching, panting and sunbathing, which might be indicative displays of discomfort from being too hot. Regardless of the amount of pasture space that chickens had, they expressed an array of natural behaviors signifying that the small amount of pasture space provided for chickens did not hinder their welfare. Results of this study suggest that providing extra pasture space to chickens may improve their wellbeing.

**Abstract:**

Access to pasture is a main benefit of free-range broiler housing systems, yet the level of outdoor stocking density on broiler animal welfare remains unsettled. The growth, feather damage, pasture ranging and behaviors were assessed for 150 mixed-sex, slow-growing Freedom Rangers from 5 to 11 weeks of age of with free access to either a high outdoor stocking density pasture (0.5 m^2^ per bird) or a low outdoor stocking density pasture (2.5 m^2^ per bird). The probability (mean, 95% CI) of tail feather damage was greater for the high-density (23.1%, 16.3 to 31.7%) compared to the low-density group (11.9%, 7.1 to 19.3%). The percent of observations resulting in sunbathing and aggressive attacks (i.e., pecking and fighting behaviors) were greater for the high-density (1.0%, 0.6 to 1.8% and 0.5%, 0.2 to 1.3%, respectively) compared to the low-density group (0.3%, 0.1 to 0.7% and 0.1%, 0.0 to 0.4%, respectively). Furthermore, an interaction between treatment and age indicated that birds in the high-density group displayed greater stretching (during weeks 7 to 10) and panting (during weeks 6 and 9). Results of this study suggest that additional outdoor pasture space may be positively associated with broiler welfare.

## 1. Introduction

Management factors, such as range enrichment provisions, have been explored as methods to improve the health and behaviors of free-range, meat-type chickens (i.e., broilers). For example, Fanatico et al. [1] reported that outdoor structural enrichments improved range utilization and decreased sitting behaviors in broilers. Dawkins et al. [2] similarly reported that broilers preferred to range in spaces which provided tree cover. Bosco et al. [3] found that olive trees and tall grass in the outdoor area encouraged broilers to range and ingest more pasture contents compared to an uncovered outdoor area. Jones et al. [4] also found that outdoor areas planted with sapling trees improved broiler ranging. These studies provide evidence that the quality of the outdoor area is important for free-range boilers, however it is unclear whether simply providing additional outdoor space for ranging improves the welfare of broilers.

It is important to understand the impact that the amount of outdoor space (i.e., outdoor stocking density) has on broiler welfare since many animal welfare programs require certain outdoor space allowances for poultry in order to meet certification labels (e.g., “free-range” labels) [5]. In fact, the amount of outdoor area provided for birds is one of the major defining characteristics differentiating between levels of these labels. The topic of outdoor stocking density is also at the forefront of organic poultry policy change in the USA since outdoor space requirements for organic poultry are currently undefined. Although the amount of pasture space allowance is a main feature of free-range poultry housing systems, the role that outdoor stocking density plays on poultry health and behavior is still not well understood.

The aim of this study was to compare the effects of two common levels of outdoor stocking densities on the growth, feather damage, pasture ranging and behaviors of free-range broilers from 5 to 11 weeks of age. The outdoor stocking densities chosen for this study were similar to the current standards for “free-range” and “pasture-raised” chickens under the Certified Humane program (Humane Farm Animal Care, Middleburg, VA, USA) and the American Humane Certified program (American Humane, Washington, DC, USA).

## 2. Materials and Methods

### 2.1. Animal Care and Housing

The University of Minnesota Institutional Animal Care and Use Committee approved all animal care and procedures specific to this experiment (protocol number #1607-33960A).

The experiment was conducted from July to October 2018 at the West Central Research and Outreach Center (Morris, MN, USA) on organic pastureland that housed organic dairy cows (*Bos taurus* L.). Details on farm management and animal care are described by Phillips et al. [6] and are therefore only briefly described in this article.

### 2.2. Experimental Design

This study was a randomized complete block design with repeated measures to evaluate 150 Freedom Ranger (Welp Hatchery, Bancroft, IA, USA) chickens (*Gallus gallus domesticus* L.) in three mixed-sex replicated groups of 50 that hatched on 29 May, 9 July and 16 July, respectively. At 4 weeks of age, birds in each replicate were leg-banded with numbered ZBands (Chicken Hill Poultry, Horseshoe Bend, ID, USA) and randomly assigned to a pen corresponding to one of two outdoor stocking density treatment groups: (1) 0.5 m^2^ of outdoor area per bird (high-density) or (2) 2.5 m^2^ of outdoor area per bird (low-density). Treatments were balanced by sex and initial body weight. The assessment of sex at 4 weeks of age had an average accuracy of 90% and was therefore not perfectly balanced between high-density (females = 33, males = 44) and low-density (females = 41, males = 32) treatment groups. The average body weights (±SD) of females and males at 4 weeks of age were 0.86 ± 0.2 kg and 0.97 ± 0.2 kg, respectively; and the average body weights of birds in the high- and low-density treatment groups were 0.91 ± 0.2 kg and 0.92 ± 0.2 kg, respectively. Birds remained in their treatment groups for the remainder of their production cycle until they reached 12 weeks of age when they were slaughtered.

Treatment pens are displayed in a photograph in Figure 1. Each pen housed 25 birds that had access to 1.8 × 3.7 m of a floorless mobile shelter (Chicken Ranger Coops, Narvon, PA, USA); thus, the covered shelter stocking density was 0.27 m^2^ per bird. Birds were confined to the shelter at night but had free access to pasture corresponding to their stocking density treatment group during the day. Birds had ad libitum access to water from an 18.2-L poultry waterer (Item # PPF5, Miller Manufacturing, Eagan, MN, USA) and granite grit from a round ground feeder (Item # PH-100, Stromberg’s, Hackensack, MN, USA). Fanatico et al. [1] reported that from 3 to 11 weeks of age free-range, mixed-sex Delaware broilers of a slow-growing genetic strain consumed an average of 138 g of concentrate per bird when feed was offered ad libitum. Furthermore, Rivera-Ferre et al. [7] calculated that a 10% restricted diet providing 115 g of concentrate per bird from 4 to 11 weeks of age was adequate for free-range broilers of a similar hybrid genetic strain (ISA) to the Freedom Ranger strain used in the present study. Based on this information, each bird received on average 141 g of concentrate (20% crude protein; Chick Starter AMP, Vita Plus Corporation, Madison, WI, USA) daily prior to shelter confinement. For each pen, the feed was placed in a 121.2-cm long galvanized steel ground trough (Item # PH-118, Stromberg’s, Hackensack, MN, USA), which was removed and sanitized the following morning.

The mobile shelter and corresponding pens were relocated every 3 to 4 days to give birds at least 50% forage ground cover. Forage biomass and height in pens were measured using a rising plate meter (30 samples per pen; Jenquip, Feilding, New Zealand) prior to and after rotation to quantify the available and residual forage, respectively. The average ± SD forage biomass throughout the study was 1.8 ± 0.5 and 1.7 ± 0.3 Mg/ha for pens of the high- and low-density treatment groups, respectively. The average ± SD forage height measured over the course of the study was 9.1 ± 3.8 and 8.7 ± 2.2 cm for pens in the high- and low-density treatment groups, respectively. The orientation of the shelter alternated between facing either East or West approximately every 3 rotations.

### 2.3. Data Collection

Body weight and feather damage scores for each individual bird was assessed prior to study initiation, starting at 4 weeks of age and weekly thereafter. Feather damage scores for the back, thigh, tail and wing areas were conducted using a visual feather assessment: 0 = fully feathered, 1 = rough, 2 = some broken feathers and small bald areas, 3 = heavily broken feathers and some bald areas, 4 = almost bald or large bald areas and 5 = bald with no feather cover [8].

Behavior observations were recorded by a single observer four times per week in the morning (between 08:00 and 11:45 h) and afternoon (between 12:00 and 18:45 h) when there was no precipitation. The time range for observations was intended to encompass time points relative to daylight between 2 h after sunrise and 2 h prior to sunset. Noon (12:00 h) was used to delineate between morning and afternoon time of day categories, therefore the range the time period of the afternoon was greater than the time period of the morning. Prior to behavior observations, pasture ranging was recorded for each pen as the number of birds outside of the shelter. Behaviors were then recorded in continuous 60-s observation periods on 10 individual focal birds per pen using the Animal Behaviour Pro mobile app (version 1.2) [9]. Focal birds were identified using livestock paint prior to study initiation and were observed in random order alternating between treatment pens. Behavioral states corresponding to the time budget were recorded as durations and behavioral events were recorded as binary outcomes (i.e., the occurrence of a behavioral event within the 60-s observation period was recorded as either a yes [presence] or no [absence]). An ethogram defining recorded behaviors is in Table 1.

The University of Minnesota West Central Research and Outreach Center weather station recorded ambient humidity, ambient temperature, precipitation, solar radiation and wind speed every 15 min. The comprehensive climate index (CCI; i.e., apparent temperature) was calculated based on ambient humidity, ambient temperature, solar radiation and wind speed [13]. For each behavioral observation, the time was rounded to the nearest 15-min and matched with the weather data.

### 2.4. Statistical Analysis

All analyses were performed in RStudio (version 1.3.1073) [14] with linear mixed models and mixed logistic regression models using the *glmmTMB* function [15]. For all models, fixed effects were replicate (3 levels), treatment (2 levels), age (7 levels), the treatment and age interaction and the random effect of experimental unit (pen; 6 levels). The first order autocovariance structure was used to account for repeated measures. Likelihood ratio tests (LRT) were used to assess the significance of fixed effects by comparing full and reduced models [16].

The analyses of body weight, feather damage and behaviors included a fixed effect of sex (2 levels) and a random effect of bird identification (ID). The analysis of body weight also included a continuous covariate for initial body weight recorded prior to treatment initiation when birds were 4 weeks of age. The interaction between treatment and sex for the analysis of body weight was initially tested but was removed from the model based on its insignificant effect. Since recorded feather damage scores were no greater than 1, feather damage scores were dichotomized into no damage (0; score = 0) and damage (1; score ≥ 1) binary outcomes and the analyses were performed under a binomial error distribution. Pasture ranging and behavior outcomes were aggregated into weekly summations; ranging and behavioral states were analyzed with a beta-binomial error distribution and behavioral events were analyzed with a binomial error distribution. No birds were observed panting during weeks 10 and 11 of the study so these weeks were removed from the analysis. Data for rarely observed behavioral events (drinking, flapping, sunbathing, aggressive display, dustbathing and aggressive attack) were pooled over weeks by obtaining a single summation for each focal bird and outcome. Behavioral events pooled across weeks did not include fixed or random effects containing age and did not include the random effect of bird ID.

Significance was declared when *p* ≤ 0.05. The Tukey adjustment was applied for pairwise comparisons. Marginal means and 95% CI for feather damage, pasture ranging and behaviors are reported as values back-transformed from the logit scale.

## 3. Results

### 3.1. Weather

Daily weather conditions while birds of all 3 replications were housed in mobile shelters (26 June to 8 October) over the course of the study are presented in Figure 2. The averages ± SD for ambient humidity, ambient temperature, solar radiation, wind speed and CCI during the study were 70 ± 9%, 20 ± 6 °C, 418 ± 174 W/m^2^, 1.1 ± 0.4 m/s and 25 ± 8 °C, respectively. On average, morning observations had 76 ± 7% humidity, 18 ± 6 °C ambient temperature, 382 ± 188 W/m^2^ solar radiation, 1.0 ± 0.4 m/s wind speed and 23 ± 8 °C CCI. Meanwhile, afternoon observations had 64 ± 8% humidity, 22 ± 6 °C ambient temperature, 454 ± 150 W/m^2^ solar radiation, 1.1 ± 0.5 m/s wind speed and 27 ± 8 °C CCI, on average.

### 3.2. Body Weight

Growth rates were similar among treatment groups, such that the mean body weights (95% CI) for broilers in the high- and low-density treatment groups were 2.2 kg (2.2 to 2.3 kg) and 2.2 kg (2.1 to 2.2 kg), respectively, when averaged across age. An effect of age indicated that birds became significantly heavier each week (Table 2). Birds gained between 0.2 and 0.4 kg per week. Furthermore, there was an effect of sex on body weight, in which the mean body weight of males was greater than females (Table 3).

### 3.3. Feather Damage

There was an effect of stocking density treatment on tail feather damage (Table 4). Broilers in the high-density group had greater tail feather damage compared to broilers in the low-density group. Yet, birds had similar wing feather damage over the course of the study regardless of treatment. There was an effect of age on wing and tail feather damage (Table 2). In general, the probability of observing wing feather damage increased as birds aged and tail feather damage was the lowest at 5 weeks of age compared to all other weeks. There was an effect of sex on wing feather damage, in which females had a greater probability for wing feather damage compared to males (Table 3). Neither back nor thigh feather damage was observed for any birds during the study.

### 3.4. Behaviors

#### 3.4.1. Pasture Ranging

There was no effect of treatment on pasture ranging, such that a similar percentage of birds were observed pasture ranging between high-density (32.7%, 95% CI = 24.2 to 42.5%) and low-density (41.6%, 95% CI = 32.0 to 52.0%) groups. There was an effect of age (Table 2) and time of day (Table 5) on pasture ranging. In general, ranging increased with age and more birds were observed ranging in the morning compared to the afternoon.

#### 3.4.2. Behavioral States

Behavioral states of the time budget were similar among treatment groups and sex. Sitting, standing and sleeping were the most commonly observed behavioral states, followed by walking and running. There was an effect of age (Table 2) and time of day (Table 5) on all behaviors of the time budget. Older birds generally had a more active time budget, in which sitting decreased and standing increased with age. Sleeping increased until 7 weeks and decreased thereafter. Walking decreased until weeks 7 to 9 and then increased thereafter. Running was greatest at weeks 5, 10 and 11 compared to week 7. A more active time budget was observed in the morning compared to the afternoon, such that more time was spend standing, walking and running during the morning and more time was spent sitting and sleeping during the afternoon.

#### 3.4.3. Behavioral Events

The behavioral events in order from greatest to least commonly recorded were: preening, foraging, stretching, grooming, disturbing, panting, drinking, flapping, sunbathing, aggressive display, dustbathing, aggressive attack. For the aggressive display category, threats and chases were most common, followed by standoffs and leaps. For the aggressive attack category, pecking was more commonly observed than fighting. There were no effects of treatment, age, sex nor time of day on drinking and dustbathing; the overall mean percentage (95% CI) of observations recorded for these behavioral events were 1.9% (1.5 to 2.6%) and 0.3% (0.1 to 0.8%), respectively.

There was an effect of stocking density treatment on two behavioral events (Table 4). Birds in the high-density group were more commonly observed sunbathing and performing aggressive attacks compared to birds in the low-density group. There was also a trend (*p* = 0.08) for the effect of treatment on foraging, such that the mean percent (95% CI) of observations in which foraging was recorded was 23.8% (20.7 to 27.3%) for birds in the low-density group and 19.3% (16.5 to 22.4%) for birds in the high-density group.

There was an interaction present between treatment and age for stretching (Figure 3). Birds in the high-density group showed greater stretching during weeks 8, 9 and 10 compared to birds in the low-density group (*p* ≤ 0.03). For birds in the low-density group, stretching was greater during week 6 compared to week 9 (*p* = 0.03) while the remaining weeks were similar. Stretching events were similar between all weeks for birds in the high-density group (*p* > 0.94).

There was an interaction present between treatment and age for panting (Figure 4). No focal birds were observed panting during weeks 10 and 11 of age. Birds in the high-density group showed greater panting during weeks 6 and 9 compared to birds in the low-density group (*p* ≤ 0.03). For birds in the low-density group, panting was greater during week 6 compared to weeks 5, 10 and 11 (*p* < 0.01) while the remaining weeks were similar. For the high-density group, panting was highest during weeks 6 and 9 compared to all other weeks (*p* ≤ 0.01).

There was an effect of age on preening, foraging, grooming and disturbance events (Table 2). The remaining behavioral events (except stretching and panting) were pooled over weeks and therefore the effect of age could not be assessed. Preening generally decreased with age. Foraging was highest during week 5 and lowest during weeks 7 and 9. In general, foraging was quite variable across age. Grooming and disturbance behaviors decreased with age.

There was an effect of sex on preening, panting, aggressive display and aggressive attack (Table 3). Preening and panting were more commonly observed for females, while aggressive displays and attacks were more commonly observed for males.

There was an effect of time of day on foraging, stretching, panting, flapping and aggressive attack events (Table 5). Foraging and aggressive attacks were more commonly observed in the morning, while stretching, panting and flapping were more commonly observed in the afternoon.

## 4. Discussion

### 4.1. Effects on Growth and Activity

There were no effects of treatment on body weight nor behavioral states in feed-restricted broilers, suggesting that growth and activity levels were not affected by outdoor stocking density. Other studies similarly reported that outdoor enrichment provisions did not affect body weight growth of free-range broilers [1,3]. Another study by Jones et al. [4] also found no effect of outdoor stocking density (1.2 vs. 2.5 m^2^ per bird) on free-range broiler growth, pasture ranging nor behaviors (i.e., drinking, foraging, lying, sleeping, standing and walking).

Body weight could conceivably be affected by stocking density if the activity or stress levels of birds are altered. For example, Sanchez-Casanova et al. [17] found that broilers raised indoors at a low stocking density (0.2 m^2^ per bird) had an increased growth, whereas broilers raised with outdoor access had decreased growth which could be partially explained by elevated activity.

### 4.2. Effects on Feather Damage and Aggression

Tail feather damage and aggressive attacks (i.e., pecking and fighting) were greater for birds reared at a high outdoor stocking density compared to a low outdoor stocking density. Gocsik et al. [18] similarly reported that plumage cleanliness was improved for broilers with a lower outdoor stocking density (1 vs. 4 m^2^ per bird). Nicol et al. [19] also reported increased feather damage and pecking behaviors as stocking density increased; however, this study investigated laying hens at indoor stocking densities (0.03 to 0.17 m^2^ per bird) much higher than those of the current study. On the contrary, Huo and Na-Lampang [20] reported no effects of indoor stocking density (0.06 to 0.13 m^2^ per bird) on aggressive attack behaviors nor feather damage in Thai crossbred broilers from 4 to 12 weeks of age.

Increased tail feather damage in birds of the high outdoor stocking density group may be partially explained by elevated aggressive attacks. An aggressive attack was a result of physical conflict between birds, such as pecking or fighting. These physical altercations were mostly comprised of pecking while fighting was rarely observed. Pecking behaviors have been previously demonstrated as a result of competition for a limited resource, such as food [21]. Although birds of the present study were feed-restricted, the amount provided was greater than amounts used in previous studies [1,7]. Alternatively, it is possible that the physical aggressive attacks were an attempt to form a social hierarchy, as suggested by Rushen [22] who reported that dominance relationships formed around 4 to 5 weeks of age in Rhode Island Red × White Leghorn pullets.

The effects of stocking density on aggression in free-range broilers is not a well understood topic. Introducing a complex environment, such as pasture access, may shift the behavioral dynamics of poultry [17,23]. Fanatico et al. [1] reported that aggressive behaviors were more likely to occur outdoors than indoors for free-range broilers. It is possible that birds in the low outdoor stocking density group were able to evade escalading aggressive conflicts by temporarily dispersing among the pasture. Meanwhile, birds in the high outdoor stocking density group may have been incapable of avoiding conflict given the limited availability of pasture space, resulting in increased physical aggressive attacks with other birds.

The occurrence of aggression was lower than reported for previous studies. For the current study, at least one of the six recorded aggression events (i.e., peck, threat, chase, standoff, fight or leap) was observed in 1.3% of observations. Fanatico et al. [1] reported that aggression events for broilers reared with access to pasture were observed in 5.4% of observations when averaged across pen location and age; however, disturbance was also categorized as aggression in this study. Regardless, aggressive displays and aggressive attacks were rarely observed compared to all other behavioral events recorded for birds of the current study.

### 4.3. Effects on Pasture Ranging

It is not surprising that outdoor stocking density did not affect pasture ranging. Previous studies demonstrated that broilers rarely venture further than the immediate vicinity of the shelter, even when provided covered areas in the outdoor space [1,2,7,24]. Although birds in the low outdoor stocking density group had more pasture space available, it is likely that they remained in close proximity to the shelter.

The average percentage of birds outside of the shelter was 37%, when aggregated across all effects. This finding is similar to Stadig et al. [24] who reported an average of 40% of broilers observed pasture ranging. However, findings for pasture ranging were greater than several previous studies, which reported that 5 to 15% of broilers were observed outside their shelters on average [1,2,4,25]. The high level of pasture ranging for the current study may have been due to differences in bird genetics and weather conditions between studies [24,26,27]. The high use of the range in the morning is in agreement with other studies [1,2,4,26,28].

### 4.4. Effects on Behaviors

In addition to aggressive attacks, outdoor stocking density also had an effect on stretching, panting and sunbathing behavioral events. Birds in the high outdoor stocking density group were more commonly observed stretching (during weeks 7 to 10), panting (during weeks 6 and 9) and sunbathing. In a study of broilers reared in a free-range system, Gonçalves et al. [29] reported that behaviors defined as “movements to stretch the wings and legs on the same side of the body simultaneously, shaking and whirring feathers, lifting part of both wings close to the body or extend the tips of the wings and/or flapping it” were more evident for fast-growing genetic strains with greater body weights and at higher temperatures, indicating that these types of behavioral adjustments may have been used to cope with discomfort, especially heat stress. Furthermore, previous studies [12,30] showed that broilers will pant as method to cope with air temperatures above their thermal neutral zone. This information possibly indicates that birds of the high outdoor stocking density experienced greater discomfort from heat compared to birds of the low outdoor stocking density group as indicated by elevated stretching and panting.

The increased sunbathing observed in broilers of the high outdoor density treatment group may also be related to high temperatures. Duncan et al. [31] suggest that radiant heat and light may trigger sunbathing in hens, which can shift to dustbathing if environmental factors are present, such as dry soil. Sunbathing and dustbathing have several shared body postures, such as side lying and feather spreading, which makes it convenient for a sunbathing bird to proceed with dustbathing. Yet, sunbathing is not a well understood behavior in domesticated poultry species and is therefore challenging to deduce the motivation for birds in the high outdoor stocking density group to perform this behavior.

The explanation for heat stress in birds of the high outdoor stocking density group remains obscured. Even though treatment groups had a similar quantity of shade from the covered shelter, it is possible that birds in the high outdoor stocking density group experienced restricted options for shade from vegetative cover, whereas birds in the low outdoor stocking density group may have had an increased opportunity to seek shade in forages due to the greater amount of space they were provided. Dense, tall stands of vegetation could theoretically provide adequate shade for free-range broilers. For example, Jones et al. [4] reported that sapling trees with a mean height of 83 cm encouraged broilers to use the range on sunny days, indicating that the vegetation in this study may have provided some relief from solar intensity in the range. This study (ibid) also found that broilers were more likely pant inside their shelter compared to in their range, suggesting that birds were able to alleviate some heat stress by seeking relief in the range.

Although the maximum average forage height in pens was only 23 cm during the current study, there was significant variation in forage height and density within pens that created a diverse habitat. Dawkins et al. [2] used preference testing to demonstrate that free-range broilers actively selected their habitat within the outdoor space provided to them, wherein birds chose habitats occupied by trees, bushes, hedge or long grass. Furthermore, tunneling behaviors in tall grasses have been documented in free-range broilers [32], which may use this adapted behavior as a method to self-regulate body temperature. Although behavioral interactions with vegetation in the outdoor area were not intentionally recorded and analyzed, birds of the current study were observed tunneling in forages and commonly used the tunnels as a place to rest. Based on this information, it is possible that the low outdoor stocking density used for this study provided birds an opportunity to seek and select a suitable habitat within their range given that they had more outdoor space than birds in the high outdoor stocking density group.

It is unclear why panting was not observed during weeks 10 and 11 of the study. Although other heat-induced behaviors were not recorded for this study, it is possible that older birds learned to cope with heat stressors by using different strategies other than panting, such as opening their wings to dissipate heat [12,33]. However, a more probable explanation is that panting was not induced due to lower air temperature during this period. Santos et al. [12] reported that panting occurred in 4 to 6 week old naked neck broilers once the average air temperature reached approximately 34 °C. For the current study, the observations for weeks 10 and 11 occurred between 17 September and 1 October, in which the maximum air temperature only reached 25 °C (Figure 2). Based on this information, it is likely that panting was not observed during weeks 10 and 11 of the study due to cooler temperatures.

Observed behaviors were modified according to time of day, which may have been due to heat stress. For the current study, sitting, sleeping, stretching, panting and flapping events were mostly observed in the afternoon when temperatures were higher, indicating that these behaviors may have been attributed to coping with heat stressors. Gonçalves et al. [29] similarly reported greater sitting, stretching and flapping and a reduction in foraging and aggressive attacks (i.e., pecking) in the afternoon. Furthermore, previous studies [12,33] also reported that broilers were more likely to exhibit panting in the afternoon. The adaption of behaviors throughout the day may have been a result of heat stress and conservation of energy in the afternoon [12,21].

The behaviors exhibited by birds was modified according to age, yet the behavioral repertoire remained diverse throughout the study, which is in agreement with previous studies that investigated the behaviors of free-range broilers. Previous studies [25,34] similarly found that pasture ranging increased with age, which may be attributed to the familiarization of the outdoor environment to birds over time. Likewise, previous research [1,35] also reported a generally more active time budget as broilers aged. A study by Gonçalves et al. [29] reported that preening, foraging and stretching (combined with flapping) were the most commonly observed behavioral events in free-range broilers, while aggression and dustbathing were rarely observed. Fanatico et al. [1] reported that foraging was the most observed behavioral event in free-range broilers, followed by drinking, preening and dustbathing. In general, the behaviors examined in the present study demonstrated a wide range of activities that free-range slow-growing broilers partook in.

### 4.5. Limitations

There is no clear indication resulting from this study that shows outdoor stocking density to be a major influencer of broiler welfare. It is likely that stocking density, regardless of whether it is indoor or outdoor, is less important than the condition of the space provided, as suggested by Dawkins et al. [36]. Hence, the results of this study are most useful when applied to production systems of similar conditions, wherein the range consists of forages with varying heights and densities, is uncovered and does not provide outdoor enrichment. Other management factors such as indoor stocking density may also play an important role in the effects of varying levels of outdoor stocking density on the welfare of broilers. A previous study reported that broiler behavior depended on both outdoor access and indoor stocking density [17], suggesting that a different covered shelter stocking density than the one used in the present study may yield different results. Furthermore, recent research suggests that future studies on free-range broiler welfare should include detailed documentation on pasture use, such as number of visits and distances traveled, as it may help predict the welfare of broilers in free-range systems [28]. Authors of the current study also suggest that future studies should perhaps investigate the complex interactions birds have with their environment such as sunbathing, tunneling and other forage manipulation behaviors, as these behaviors are presently not well understood but are presumably fundamental behaviors in free-range systems.

## 5. Conclusions

Assessing the performance and behaviors of free-range broilers from 5 to 11 weeks of age provided evidence that additional outdoor pasture space may be positively associated with broiler welfare, including reduced tail feather damage, aggressive attacks and behaviors akin to discomfort such as stretching, panting and sunbathing. Furthermore, these findings also demonstrate the extensive array of species-specific behaviors of broilers raised in a free-range system. Results from this study suggest that the level of outdoor stocking density may play a role in improving free-range broiler welfare.

## Figures and Tables

**Figure 1 animals-11-00688-f001:**
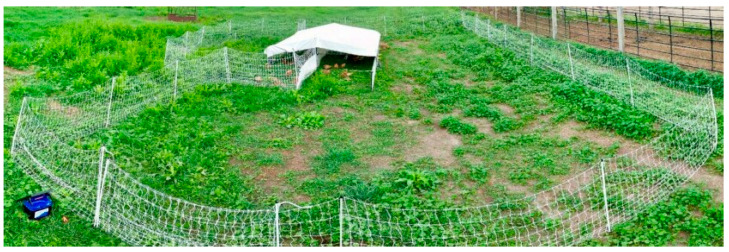
View of high (**left**) and low (**right**) outdoor stocking density treatment pens for replicate 1 birds on 27 June at 08:55. At the time this photo was taken, the average forage biomass for high- and low-density pens was 2.6 and 1.6 Mg/ha, respectively; and the average forage height for high- and low-density pens was 15.2 and 7.5 cm, respectively.

**Figure 2 animals-11-00688-f002:**
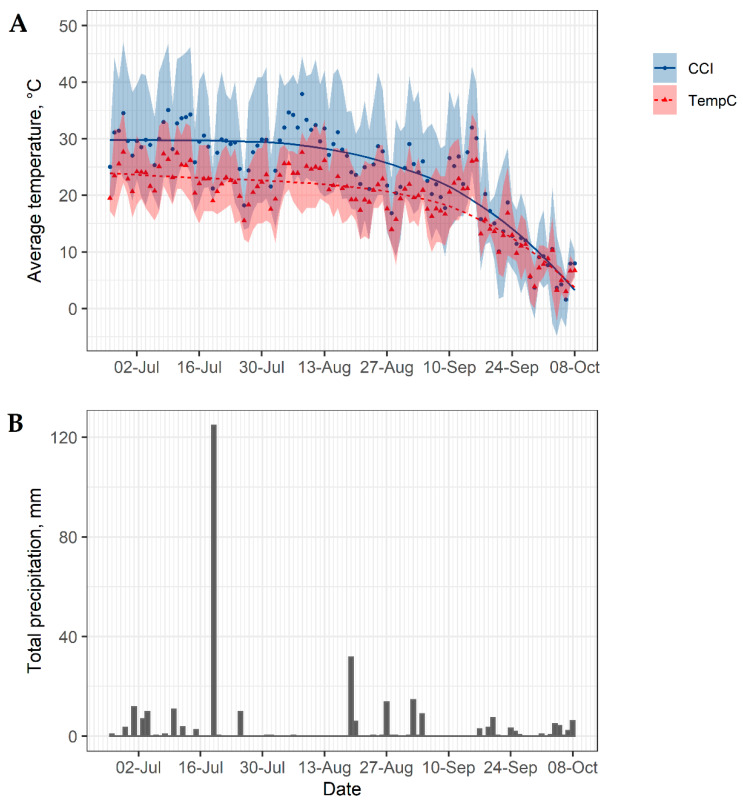
(**A**) Average daily ambient temperature and average daily comprehensive climate index (CCI; i.e., apparent temperature). Lines are the best fit locally estimated scatterplot smoothing (loess) regressions. Transparent bands represent daily minimum and maximum values. (**B**) Total daily precipitation.

**Figure 3 animals-11-00688-f003:**
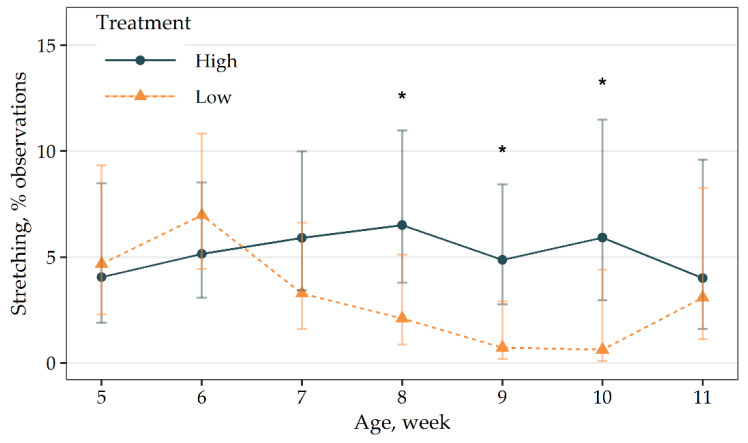
Means (± 95 CI) of stretching events for outdoor stocking density treatment (high: 0.5 m^2^ of pasture per bird; low: 2.5 m^2^ of pasture per bird) and age interaction effect. * = means within a week are different after Tukey’s adjustment, *p* ≤ 0.03.

**Figure 4 animals-11-00688-f004:**
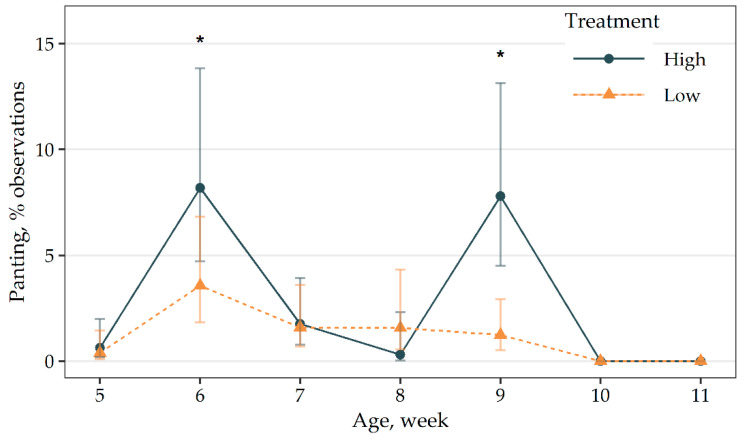
Means (±95 CI) of panting events for outdoor stocking density treatment (high: 0.5 m^2^ of pasture per bird; low: 2.5 m^2^ of pasture per bird) and age interaction effect. No birds were observed panting during weeks 10 and 11 so these weeks were removed from the analysis. * = means within a week are different after Tukey’s adjustment, *p* ≤ 0.01.

**Table 1 animals-11-00688-t001:** Ethogram for behaviors of mixed-sex Freedom Ranger chickens raised in a free-range system from 5 to 11 weeks of age observed in the range and shelter. Behaviors and descriptions are modified from Ventura et al. [10], Mollenhorst et al. [11] and Santos et al. [12].

	Description
Behavioral states ^1^	
Sitting	Bird has its breast in contact with the ground. Eyes are open
Standing	Bird maintains upright position on its extended, stationary legs
Sleeping	Bird has its breast in contact with the ground. Eyes are closed
Walking	Bird moves across the ground, wherein the legs propel the bird at a low speed
Running	Bird moves across the ground, wherein the legs propel the bird at a high speed
Behavioral events ^2^	
Preening	Bird uses its beak to peck, stroke or comb plumage
Foraging	Bird pecks or scratches at the ground
Stretching	Bird elongates its wing or its leg slowly
Grooming	Bird cleans, massages or rubs itself using beak or feet
Disturbance	A bird makes physical contact with a resting bird, causing it to adjust or stand
Panting	Bird has beak open to respire
Drinking	Bird submerges beak into the water of the drinker
Flapping	Bird is in an upright position and extends its wings repeatedly
Sunbathing	Bird holds one or both wings out from the body with feathers spread
Dustbathing	Lying bird tosses dirt onto its back and wings by ruffling and shaking its body
Aggressive attack ^3^	
Peck	Bird raises its head and strikes another bird with its beak
Fight	Two standing birds raise heads to face each other, one or both deliver > 2 kicks to opponent
Aggressive display ^4^	
Threat	Bird stands with raised feathers and neck while opponent holds its head at lower level
Chase	A bird runs > 3 steps after another bird
Standoff	Two birds face each other with heads at same level for more > 2 s
Leap	Two birds face each other, one or both jump without extending legs toward other bird

^1^ Behavioral states are mutually exclusive. Recorded as duration; ^2^ Behavioral events are non-mutually exclusive and behavioral events and states are non-mutually exclusive. Recorded as binary outcomes. ^3^ Observations in the categories peck and fight were analyzed as aggressive attack. ^4^ Observations in the categories threat, chase, standoff and leap were analyzed as aggressive display.

**Table 2 animals-11-00688-t002:** Means ± 95% CI of animal-based indicators and behaviors affected by age for mixed-sex Freedom Ranger chickens raised in a free-range system from 5 to 11 weeks of age ^1^.

	Age (Weeks)							Age Effect ^2^	
Animal-Based Indicators and Behaviors	5	6	7	8	9	10	11	*Χ* ^2^ _(6)_	*p*
Body weight, kg	1.1 (1.1, 1.2) ^g^	1.5 (1.4, 1.5) ^f^	1.9 (1.8, 1.9) ^e^	2.3 (2.2, 2.3) ^d^	2.5 (2.5, 2.6) ^c^	3.0 (2.9, 3.0) ^b^	3.2 (3.1, 3.2) ^a^	1872.7	<0.01
Wing feather damage, % probability	23.9 (13.7, 38.3) ^d^	55.8 (43.4, 67.5) ^c^	73.2 (60.9, 82.7) ^b,c^	79.2 (67.4, 87.5) ^b^	82.7 (71.5, 90.5) ^b^	96.4 (91.1, 98.6) ^a^	94.8 (88.5, 97.8) ^a^	139.1	<0.01
Tail feather damage, % probability	2.6 (0.7, 9.0) ^b^	17.3 (10.1, 27.8) ^a^	28.2 (18.7, 40.1) ^a^	25.5 (16.5, 37.3) ^a^	21.8 (13.6, 33.2) ^a^	19.0 (11.4, 29.9) ^a^	21.6 (13.3, 33.0) ^a^	29.4	<0.01
Ranging, % birds	34.2 (23.2, 47.2) ^b,c^	17.7 (10.6, 27.9) ^c^	25.9 (16.7, 37.8) ^b,c^	36.2 (24.8, 49.4) ^b,c^	38.0 (26.7, 50.8) ^b,c^	44.6 (32.2, 57.7) ^a,b^	69.1 (51.7, 82.4) ^a^	22.1	<0.01
Sitting, % time	55.0 (47.4, 62.4) ^a^	52.3 (44.7, 59.7) ^a,b^	49.4 (41.9, 57.0) ^a,b^	45.6 (38.3, 53.2) ^a,b^	43.2 (36.0, 50.7) ^b^	16.6 (12.1, 22.4) ^c^	15.8 (11.3, 21.6) ^c^	128.1	<0.01
Standing, % time	22.8 (17.3, 29.4) ^b,c^	21.4 (16.3, 27.6) ^c^	15.2 (11.2, 20.3) ^c^	23.8 (18.2, 30.4) ^b,c^	32.0 (25.4, 39.5) ^b^	64.1 (55.1, 72.3) ^a^	56.3 (47.0, 65.2) ^a^	134.0	<0.01
Sleeping, % time	10.4 (7.8, 13.7) ^c,d^	22.6 (18.2, 27.6) ^a^	26.5 (21.6, 31.9) ^a^	18.5 (14.6, 23.2) ^a,b^	13.3 (10.3, 17.1) ^b,c^	7.1 (4.8, 10.4) ^c,d^	5.0 (3.2, 7.7) ^d^	89.4	<0.01
Walking, % time	3.0 (2.2, 4.2) ^a,c^	1.7 (1.2, 2.5) ^b,d^	1.5 (1.0, 2.2) ^d^	1.3 (0.9, 2.0) ^d^	1.5 (1.0, 2.2) ^d^	1.6 (1.0, 2.6) ^c,d^	3.4 (2.3, 5.1) ^a,b^	29.4	<0.01
Running, % time	0.2 (0.1, 0.4) ^a^	0.1 (0.0, 0.2) ^a,b^	0.0 (0.0, 0.1) ^b^	0.1 (0.0, 0.2) ^a,b^	0.1 (0.0, 0.2) ^a,b^	0.2 (0.1, 0.4) ^a^	0.2 (0.1, 0.4) ^a^	27.7	<0.01
Preening, % observations	29.4 (24.6, 34.8) ^a,b^	30.1 (26.4, 34.1) ^a^	24.9 (21.0, 29.2) ^a,b,c^	20.6 (16.9, 25.0) ^b,c^	25.9 (22.2, 29.9) ^a,b,c^	22.2 (17.3, 27.8) ^a,b,c^	16.1 (11.7, 21.7) ^c^	24.9	<0.01
Foraging, % observations	29.8 (24.5, 35.8) ^a^	23.9 (20.0, 28.2) ^a^	15.4 (12.1, 19.4) ^b^	20.9 (16.7, 25.9) ^a,b^	15.8 (12.6, 19.7) ^b^	24.6 (18.8, 31.4) ^a,b^	22.6 (16.6, 29.8) ^a,b^	29.5	<0.01
Grooming, % observations	7.4 (4.9, 11.0) ^a^	4.6 (3.1, 6.9) ^a,b^	2.8 (1.6, 4.9) ^a,b^	2.2 (1.1, 4.4) ^b^	2.4 (1.3, 4.4) ^b^	2.2 (0.9, 5.5) ^a,b^	2.3 (0.7, 6.7) ^a,b^	18.1	<0.01
Disturbance, % observations	6.3 (3.9, 10.2) ^a^	3.5 (2.1, 5.9) ^a,b^	2.0 (0.9, 4.3) ^b^	2.3 (1.1, 4.8) ^a,b^	1.0 (0.4, 2.3) ^b^	1.0 (0.2, 4.2) ^a,b^	3.4 (1.3, 8.4) ^a,b^	29.8	<0.01

^1^ Behaviors of drinking, flapping, sunbathing, aggressive display, dustbathing and aggressive attack were pooled over age. Stretching (*Χ*^2^
_(6)_ = 18.5, *p* < 0.01) and panting (*Χ*^2^
_(6)_ = 16.6, *p* < 0.01) were affected by the treatment × age interaction effect. ^2^ Chi-square statistic of likelihood ratio test (LRT). ^a–g^, means within a row with different letter superscripts are different after Tukey’s adjustment, *p* ≤ 0.05.

**Table 3 animals-11-00688-t003:** Means ± 95% CI of animal-based indicators and behaviors affected by sex for mixed-sex Freedom Ranger chickens raised in a free-range system from 5 to 11 weeks of age ^1^.

	Sex		Sex Effect ^2^	
Animal-based Indicators and Behaviors	Male	Female	*Χ* ^2^ _(1)_	*p*
Body weight, kg	2.3 (2.3, 2.4)	2.1 (2.0, 2.1)	73.3	<0.01
Wing feather damage, % probability	68.0 (57.5, 77.0)	86.6 (78.8, 91.9)	16.6	<0.01
Preening, % observations	20.9 (18.8, 23.3)	27.0 (24.5, 29.7)	13.2	<0.01
Panting, % observations	1.2 (0.7, 2.2)	2.0 (1.2, 3.4)	3.8	0.05
Aggressive display, % observations	1.0 (0.3, 2.7)	0.1 (0.0, 0.6)	13.3	<0.01
Aggressive attack, % observations	0.5 (0.2, 1.2)	0.1 (0.0, 0.5)	5.4	0.02

^1^ Tail feather damage (*Χ*^2^
_(1)_ = 0.5, *p* = 0.47), sitting (*Χ*^2^
_(1)_ = 0.5, *p* = 0.50), standing (*Χ*^2^
_(1)_ = 0.1, *p* = 0.70), sleeping (*Χ*^2^
_(1)_ = 2.1, *p* = 0.14), walking (*Χ*^2^
_(1)_ = 0.2, *p* = 0.67), running (*Χ*^2^
_(1)_ = 0.0, *p* = 0.99), foraging (*Χ*^2^
_(1)_ = 1.0, *p* = 0.31), stretching (*Χ*^2^
_(1)_ = 0.1, *p* = 0.80), grooming (*Χ*^2^
_(1)_ = 1.5, *p* = 0.23), disturbance (*Χ*^2^
_(1)_ = 0.4, *p* = 0.51), drinking (*Χ*^2^
_(1)_ = 0.0, *p* = 0.82), flapping (*Χ*^2^
_(1)_ = 0.7, *p* = 0.39), sunbathing (*Χ*^2^
_(1)_ = 0.1, *p* = 0.81) and dustbathing (*Χ*^2^
_(1)_ = 0.2, *p* = 0.67) were not affected by sex. ^2^ Chi-square statistic of likelihood ratio test (LRT).

**Table 4 animals-11-00688-t004:** Means ± 95% CI of animal-based indicators and behaviors affected by outdoor stocking density treatment (high: 0.5 m^2^ of pasture per bird; low: 2.5 m^2^ of pasture per bird) for mixed-sex Freedom Ranger chickens raised in a free-range system from 5 to 11 weeks of age ^1^.

	Treatment		Treatment Effect ^2^	
Animal-Based Indicators and Behaviors	High	Low	*Χ* ^2^ _(1)_	*p*
Tail feather damage, % probability	23.1 (16.3, 31.7)	11.9 (7.1, 19.3)	6.2	0.01
Sunbathing, % observations	1.0 (0.6, 1.8)	0.3 (0.1, 0.7)	5.1	0.02
Aggressive attack, % observations	0.5 (0.2, 1.3)	0.1 (0.0, 0.4)	6.9	<0.01

^1^ Body weight (*Χ*^2^
_(1)_ = 1.5, *p* = 0.22), wing feather damage (*Χ*^2^
_(1)_ = 1.0, *p* = 0.32), ranging (*Χ*^2^
_(1)_ = 1.1, *p* = 0.28), sitting (*Χ*^2^
_(1)_ = 1.0, *p* = 0.32), standing (*Χ*^2^
_(1)_ = 0.7, *p* = 0.40), sleeping (*Χ*^2^
_(1)_ = 0.2, *p* = 0.68), walking (*Χ*^2^
_(1)_ = 0.2, *p* = 0.64), running (*Χ*^2^
_(1)_ = 0.0, *p* = 0.99), preening (*Χ*^2^
_(1)_ = 0.1, *p* = 0.72), foraging (*Χ*^2^
_(1)_ = 3.0, *p* = 0.08), grooming (*Χ*^2^
_(1)_ = 0.3, *p* = 0.60), disturbance (*Χ*^2^
_(1)_ = 1.6, *p* = 0.21), panting (*Χ*^2^
_(1)_ = 2.4, *p* = 0.12), drinking (*Χ*^2^
_(1)_ = 2.0, *p* = 0.16), flapping (*Χ*^2^
_(1)_ = 0.0, *p* = 0.86), aggressive display (*Χ*^2^
_(1)_ = 0.0, *p* = 0.83) and dustbathing (*Χ*^2^
_(1)_ = 0.1, *p* = 0.82) were not affected by treatment. Stretching (*Χ*^2^
_(1)_ = 18.5, *p* < 0.01) and panting (*Χ*^2^
_(1)_ = 16.6, *p* < 0.01) were affected by the treatment × age interaction effect. ^2^ Chi-square statistic of likelihood ratio test (LRT).

**Table 5 animals-11-00688-t005:** Means ± 95% CI of behaviors affected by time of day for mixed-sex Freedom Ranger chickens raised in a free-range system from 5 to 11 weeks of age ^1^.

	Time of Day		Time of Day Effect ^2^	
Behaviors	Morning	Afternoon	*Χ* ^2^ _(1)_	*p*
Ranging, % birds	47.1 (38.6, 55.7)	28.0 (21.5, 35.6)	18.6	<0.01
Behavioral state, % time				
Sitting	31.6 (26.6, 37.2)	44.5 (38.6, 50.7)	46.8	<0.01
Standing	39.9 (34.0, 46.1)	24.8 (20.2, 30.0)	61.2	<0.01
Sleeping	11.3 (9.6, 13.2)	15.1 (13.0, 17.4)	11.0	<0.01
Walking	2.2 (1.7, 2.9)	1.6 (1.2, 2.1)	10.0	<0.01
Running	0.1 (0.1, 0.2)	0.1 (0.0, 0.1)	10.8	<0.01
Behavioral event, % observations				
Foraging	24.8 (22.0, 27.9)	18.5 (16.0, 21.2)	15.0	<0.01
Stretching	2.6 (1.8, 3.7)	4.6 (3.4, 6.3)	11.0	<0.01
Panting	0.7 (0.4, 1.2)	3.7 (2.4, 5.9)	75.6	<0.01
Flapping	0.3 (0.1, 0.8)	1.0 (0.5, 1.7)	4.8	0.03
Aggressive attack	0.4 (0.2, 1.2)	0.1 (0.0, 0.5)	5.6	0.02

^1^ Preening (*Χ*^2^
_(1)_ = 1.7, *p* = 0.19), grooming (*Χ*^2^
_(1)_ = 0.0, *p* = 0.98), disturbance (*Χ*^2^
_(1)_ = 0.1, *p* = 0.79), drinking (*Χ*^2^
_(1)_ = 0.9, *p* = 0.34), sunbathing (*Χ*^2^
_(1)_ = 0.6, *p* = 0.44) aggressive display (*Χ*^2^
_(1)_ = 1.0, *p* = 0.32) and dustbathing (*Χ*^2^
_(1)_ = 0.9, *p* = 0.33) were not affected by time of day. ^2^ Chi-square statistic of likelihood ratio test (LRT).

## Data Availability

Data generated from this experiment is available on request from corresponding authors.

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
