# Peer review of "Effects of Outdoor Stocking Density on Growth, Feather Damage and Behavior of Slow-Growing Free-Range Broilers"

_animals, 2021, doi:10.3390/ani11030688_

Round 1

Reviewer 1 Report

Dear authors,

I think the issue of your study is very important to impove broiler welfare. The comparison of diffrent outdoor stocking densities on broiler health and welfare is therefore very interessting. 

I have some minor suggestions

In general table headdings should be more precise. Maybe every headline could include the information of the animal species and a bit more information about 'outcome'. E.g. animal based indicators and behaviours instead 'outcome'.

Maybe the investiagtion would be even clearer if a distinction were made between feather pecking and aggressive pecking, because the two have different underlying causes.

Line 113 must be reference number [8] instead of [23]

Line 119/120 Headline of Table 1 could be more precise as it should be understandable without the text. E.g. 'Ethogram of broiler behaviours modified [...] observed in the free range area (or indoor?)

Line 180 Headline of Table 2 could be more precise. E.g. Means [...]  for body weight, feather damage and behaviours in % assessed in x broilers that are affected by age listed for week of age 5-11.

Lines 185, 201/202, 218 headdings more precise

Line 268 must be Figure 4 instead of Figure 3

Reviewer 2 Report

Please see my comments or remarks on MS indicated by yellow notes on the underline.

Reviewer 3 Report

Review of “Effects of outdoor stocking density on growth, feather damage 2 and behavior of slow-growing free-range broilers”

General Comments:  The manuscript compares feed limited slow-growing broiler performance and wellbeing at 2 pasture stocking densities.

The description of the housing and management is inadequate and the reference to Phillips [6} does not provide the type of watering or feeding equipment, the orientation of the covered shade area if the orientation was alternated between the two treatments with each replication.

Since the covered shade area apparently had mesh wire walls it should not be considered “indoor housing” but covered shelter.

The days of hatch and days of the observation and the number of birds to not add up.

The assumption that “stretching” is an aggressive negative animal welfare is not supported by the reference. Gonçalves et al. [28]

Specific comments:

L67 States “The experiment was conducted from July to October 2018”, L74 states “hatched on 29 May, 9 July and 13 August” and L82 states “remainder of their production cycle until they reached 12 weeks of age.”

Hatch                  4 weeks              12 weeks

05/29                  06/26                  08/21

07/09                  08/06                  10/01

09/13                  10/11                  12/06

Why is the data for Table 2, Figure 3 “Stretching” and Figure 3 “Panting” only from 5 weeks to 11 weeks not 12 weeks?

Explain was the 95% CI are not balanced above and below the mean but are not in Figure 3 “Stretching” and Figure 3 “Panting”.

L48 Provide correct reference since reference for stocking density for “Freedom Rangers” since reference [5] is a review entitled “Review of environmental enrichment for broiler chickens” and references from 2 to 22 birds per m^2 for diverse genetic stocks.

Reference [5] states “EU marketing legislation (Commission Regulation (EEC) No 1538/91) requires that free-range poultry is provided with 1 m^2 per bird and that poultry marketed as “traditional free range” (such as Label Rouge in France) is provided with 2 m^2 per bird.”

L20 states “Freedom Rangers from 5 to 11 weeks of age of with free access to either a high outdoor stocking density pasture (0.5 m^2 per bird) or a low outdoor stocking density pasture (2.5 m^2 per bird).

Therefore, both the 0.5 pens and the 2.5 m^2 pens do not fit within the EU recommended standards.

L78 Provide 4 week body weight values by gender and treatment assignment.

L80 Explain how 34 + 44 + 42 + 32 = 152 total birds but in L73 only 150 hatched chicks were obtained.

Table 1 Behavioral ethogram prepared for broilers. Behavior

Description

Sitting

Characterized by Behavior When the body of the bird is in contact with the ground, floor, or bedding

Eating

Consuming or pecking feed at the feeder

Drinking

Consuming water at the drinker

Foraging

Consuming or pecking the vegetal substrate in the paddock area

Preening

Exploring the feathering with beak for both maintenance as investigation

Non aggressive pecking

Light pecking other birds, usually in the lower ventral region of the neck, back, base and tip of the tail or abdomen

Aggressive pecking

Strong pecking another bird triggering aggressive or defensive reaction, usually directed to the upper region of the head and comb or in the dorsal lower region of the neck

Discomfort movements

Movements to stretch the wings and legs on the same side of the body simultaneously, shaking and whirring feathers, lifting part of both wings close to the body or extend the tips of the wings and/or flapping it

Scratching the ground

The bird explores his territory with his feet and beak, directed to the floor

Dust bathing

Revolving in the bedding substrate or on the soil in the paddock area, spreading it throughout the body

Heat Stress Management in Broilers. Gary D. Butcher and Richard Miles

“Panting would normally be expected to occur when the ambient temperature is near or above 30o C. Relative humidity influences evaporative heat loss through panting. Broilers, as well as other domestic poultry, cannot tolerate high temperature coupled with high relative humidity.”

L88 Provide a reference that an average of “121 g of concentrate” is adequate for Freedom Ranger broilers.

L91 When the shelter was moved, was the orientation of the covered shade area constant or rotated?

L102 State if birds were banded and individual data recorded. L141 “bird ID”.

L104 Explain why you use a feather scoring that was developed by Kretzschmar-McCluskey [7] to evaluate broiler breeder hens “test on the areas of the female that have the most contact with the males during mating, including the back, thighs, wings and tail.”

L108 Explain why the morning observation occurred over a 3 hr and 45 min and the afternoon observation period occurred over 6 hr and 45 min.

“Behavior observations were recorded by a single observer four times per week in the morning (between 08:00 and 11:45 h) and afternoon (between 12:00 and 18:45 h).

L135 States for “age (6 levels)” but in Table 2 there are 7 ages (5, 6, 7, 8, 9,10, 11).  What happened to week 12.

L167 State that in Figure 2. A.  if the solid blue and dotted red lines are best fit regressions.

L168 In Figures 3 explain why the Date runs from July 1 to October 1 but the birds were in the pens from Jun 25 (5/29 hatch + 4 wks = 6/25) to Nov 5 (8/13 hatch +12 wks = 11/05).

Stat that the data presented is for all three replications with and n=10 broilers/replication.

L172 The authors need to report 11/12 wk body weight by treatment and sex in the text and Table 2.

Explain the usefulness of mean body weight for feed restricted broilers with uneven number of males and females within the treatments.

L196 and L197 Tail feather damage only differed for 5 (b) wk compared to wks 6 thru 11(all a).

L232 Correct statement since Running did not significantly differ from wk 6 thru 9 all superscript “b”.

L235-237 Indicate which behaviors were not significantly different in the numerical list of appearance.

L243 Replace “several” with two (sunbathing and aggressive attacks L244-245).

L253 Insert “and 10”, …weeks 9 and 10…

L256 Explain why the 95% CI are not evenly spaced above and below the means.

L287 State that the broilers were feed limited.

L297 States “Furthermore, this study (ibid) found that broilers raised indoors at a high stocking density (0.1 m2 per bird) had decreased growth despite their low activity, which was attributed to high levels of stress as indicated by the heterophil to lymphocyte ratio.” But there were no differences in body weight at wk 6 between low and high density.

Chickens raised indoors were heavier on week 3 (1.09 ± 0.06 vs. 1.05 ± 0.06; p = < 0.001), 5 (2.70 ±

0.18 vs. 2.60 ± 0.19; p = 0.002) and 6 (3.39 ± 0.21 vs. 3.28 ± 0.23; p = 0.001), compared with those with

outdoor access, and those kept at low density were heavier on weeks 4 (1.90 ± 0.13 vs. 1.84 ± 0.11;

p = 0.004) and 5 (2.71 ± 0.18 vs. 2.60 ± 0.17; p = < 0.001), compared to those housed at high density,

as seen in Figures 7 and 8.

Delete statement since the difference in H/L ratio was in low density not high density.  “A significant interaction demonstrated that the H/L ratio was particularly reduced by outdoor access in birds at low density on week 6 (outdoor access 1.44 (b) ± 0.94, no outdoor access 3.21 (a) ± 0.95, p = 0.01),”

L317 Here you reference that “competition for limited resource, such as food” could impact pecking behaviors.  Therefore, justify your feed limiting management practice since it appears to induce pecking.

L343 Provide data to support statement that “diurnal rhythm or ambient temperature” were “likely” to explain “high use of the range in the morning” or delete as speculation.

L357 Gonçalves et al. [28] defined “Discomfort movements”  for stretching as “Movements to stretch the wings and legs on the same side of the body simultaneously” while you have defined stretching as “Stretching  = Bird elongates its wing or its leg slowly” with no mention of simultaneously.

“In the afternoon, most of the animals of each genetic line remained sitting and had the others behaviors greatly reduced, motivated by the increased temperature.” No mention that stretching was a response to heat stress.

L373 Delete statement as assumed speculation without data to support.

Duncan did not mention “heat stress” and it is obvious that there would be “radiant heat and light” during sunbathing.

L377 States “It is possible that birds in the high outdoor stocking density group 377 may have experienced greater heat stress due to restricted options for shade, whereas 378 birds in the low outdoor stocking density group may have had an increased opportunity 379 to seek shade in forages due to the greater amount of space they were provided.”

However, both high and low density should of had that same quantity of shade from the covered shelter and in Figure 1 indicates minimal shade in the pasture from forage.  Add the time of day that  the photograph was for Figure 1.

Round 2

Reviewer 3 Report

The authors have responded and revised the manuscript to address my comments adequately.

I have indicated a few suggested wording changes below:

L85 Suggest replacing “butchered” with “slaughtered”, …when they were slaughtered.

L95 Reword replace “90%” with “10%”, … Furthermore, Rivera-Ferre et al. [7] calculated that a 10% restricted diet providing 115 g of concentrate per bird from 4 to 11 weeks of age was adequate for 96 free-range broilers

Rivera-Ferre et al. [7] stated “restricted basis (approximately 90% of ad libitum)”.

L430 Reword since when the birds are coping with a heat stressor they are no longer subjected to “heat stress”.

…older birds learned to cope with heat stressors by using different strategies…

L442 Replace “ stress” with “stressors”.

…that these behaviors may have been attributed to coping with heat stressors.
